# Enhanced audience sentiment analysis in IoT-integrated metaverse media communication

**Hongtao Wang[1], Shan Wang[1]\*, Yijun Lu[2], Nikolai Ivanovich Vatin[3], Jiandong Huang** [2,3]\*

**1** College of Music and Dance, Guangzhou University, Guangzhou, China, **2** School of Civil and Transportation Engineering, Guangzhou University, Guangzhou, China, **3** Peter the Great St. Petersburg Polytechnic University, Saint Petersburg, Russian Federation

\* wangshan@gzhu.edu.cn (SW); jiandong.huang@hotmail.com (JH)

**Data availability statement:** The dataset of the Twitter Sentiment140 has been uploaded in the supporting information. The Amazon Customer

## Abstract

The convergence of Metaverse technologies, Internet of Things (IoT), and consumer electronics has given rise to an imperative need for scalable, real-time sentiment analysis that can process heterogeneous, high-velocity media flows. The traditional approaches tend to fail in preserving the contextual, emotional, and temporal dynamism that pervades cross-platform settings. For these shortcomings, this work proposes a deep learning-based framework for sentiment analysis that integrates IoT-enabled consumer devices and Metaverse media interactions seamlessly. The overall BG-Hybrid model, fundamentally, blends BERT-led bidirectional encoding and GPT-based generative modeling to attain subtle emotion detection and context-aware comprehending. The five interconnected modules constituting the architecture include (i) multi-source data collection using RESTful APIs; (ii) weighted preprocessing pipelines using tokenization, lemmatization, and normalization; (iii) Adam algorithm-optimized model training and cross-entropy loss minimization-based training; (iv) adaptive real-time processing using dynamic window segmentation; and (v) an ongoing refinement loop using continuous user inputs, triggered by a feedback mechanism. Predictive thresholding is employed to manage temporal sentiment variations, and anomaly detection ensures data trustworthiness. Experimental analyses on Twitter Sentiment140 and Amazon Reviews datasets validate the effectiveness of the system, obtaining 94.5% accuracy, 91.5% F1-score, an average response latency of 250 ms, and proved scalability exceeding 91.5%.

## Introduction

The integration of Metaverse technologies, consumer electronics, and the Internet of Things (IoT) is reshaping human–machine interactions [1]. Persistent connectivity through smart devices-such as wearables, home automation systems, and augmented reality platforms-enables synchronization between virtual and physical environments [2]. To support this interoperability, systems must optimize data throughput, energy efficiency, and communication latency [3,4]. However, achieving real-time sentiment analysis in these

Reviews dataset is publicly available via the AWS Registry of Open Data at: (https://registry.opendata.aws/amazon-reviews/). Amazon Customer Reviews can also be found via the reference 46 and we have properly cited in the manuscript.

**Funding:** This research was funded by the 2024 Special Research Project on People-To-People Exchange of the Center for International People-To-People Exchange and the Institute of Research and Practice of International People-To-People Exchange of the Ministry of Education (CCIPE-YXSJ-20240060 to S.W.), the Ministry of Education's Employment Education Project for Supply-Demand Matching in 2025 (2024120908493 to S.W.), the 2022 Annual Research Project: "Online Open Courses Promoting Ideological and Political Education in Courses – Practice and Reflection on Red Classics' Education through Dance" (2022ZXKC361 to S.W.), the Natural Science Foundation of Guangdong Province, China (Grant No. 2024A1515011162 to J.H.), the Natural Science Foundation of Shandong Province, China (Grant No. ZR2024QE021 to J.H.), and the Ministry of Science and Higher Education of the Russian Federation within the framework of the state assignment No. 075-03-2022-010 dated 14 January 2022 and No. 075–01568-23-04 dated 28 March 2023 (Additional agreement 075-03-2022-010/10 dated 9 November 2022, Additional agreement 075-03-2023-004/4 dated 22 May 2023 to N.V.). The funders had no role in study design, data collection and analysis, decision to publish, or preparation of the manuscript.

**Competing interests:** The authors have declared that no competing interests exist.

heterogeneous, dynamic ecosystems remains a challenge due to evolving linguistic patterns, limited cross-device compatibility, and asymmetric computational resources [5,6].

Current sentiment analysis frameworks often fall short when addressing multimodal, multilingual, and temporally dynamic data streams common in IoT-enabled Metaverse applications [7,8]. Challenges such as cultural idioms, regional dialects, and rapidly shifting online expressions further complicate the accurate interpretation of user emotions [9,10]. This paper introduces a high-fidelity, real-time sentiment analysis framework designed to adapt dynamically to diverse media data. The architecture embodies contextual, emotional, and temporal subtleties while preserving computational efficacy and scalability to achieve deployment on diverse IoT devices.

The motivations behind this research come from an increasing need to develop scaleable, reliable, and contextually-aware tools to perform sentiment analysis on new Metaverse-led platforms. Existing methods take minimal account of the need to achieve cross-platform interoperability, multilingual processing, and adaptive response routines essential to accurate sentiment detection. To address these gaps, this work offers a resilient, deep learning-based framework that facilitates precise sentiment analysis within real-time settings and that is capable of running on very different consumer end-points.

Center to the system proposal stands a new BG-Hybrid deep learning framework that encapsulates BERT-based bidirectionally contextual encoding and GPT-directed generative functionality. The architecture comprises five interrelated modules (Fig 1): (i) data gathering through RESTful API scraping to achieve multi-platform aggregation; (ii) preprocessing by performing weighted normalization, tokenization, lemmatization, and multilingual translation; (iii) training using a cross-entropy loss function optimized by the Adam optimizer with adaptive learning rates; (iv) real-time sentiment analysis using dynamic window segmentation and anomaly detection; and (v) refinement of feedback using entropy-directed parameter adjustment to achieve user-aligned prediction. This architecture allows the system to efficiently process divergent data streams, dynamically adjust to temporal changes, and repeatedly improve its performance to work within IoT-integrated Metaverse settings. The main contributions of the proposed approach can be summarized as:

1. Hybridized deep learning model combining BERT's bidirectional contextual encoding and GPT's generative sentiment understanding, tailored for real-time, fine-grained emotion analysis in heterogeneous IoT-Metaverse environments.
2. Real-time adaptive feedback mechanism that incorporates user corrections and evolving linguistic trends to iteratively refine model parameters through entropy minimization strategies.
3. Temporal segmentation strategy dynamically adapting to data stream velocities and contextual variations, enhancing real-time sentiment detection accuracy under fluctuating media traffic conditions.

The remainder of this article is organized as follows. Background & Related Work Analysis discusses the related existing approaches. Materials and methods details the Proposed Methodology, describing the design and implementation of the novel deep learning-based sentiment analysis system. Discussion comprises a discussion and outcome. The article concludes with Conclusion section.

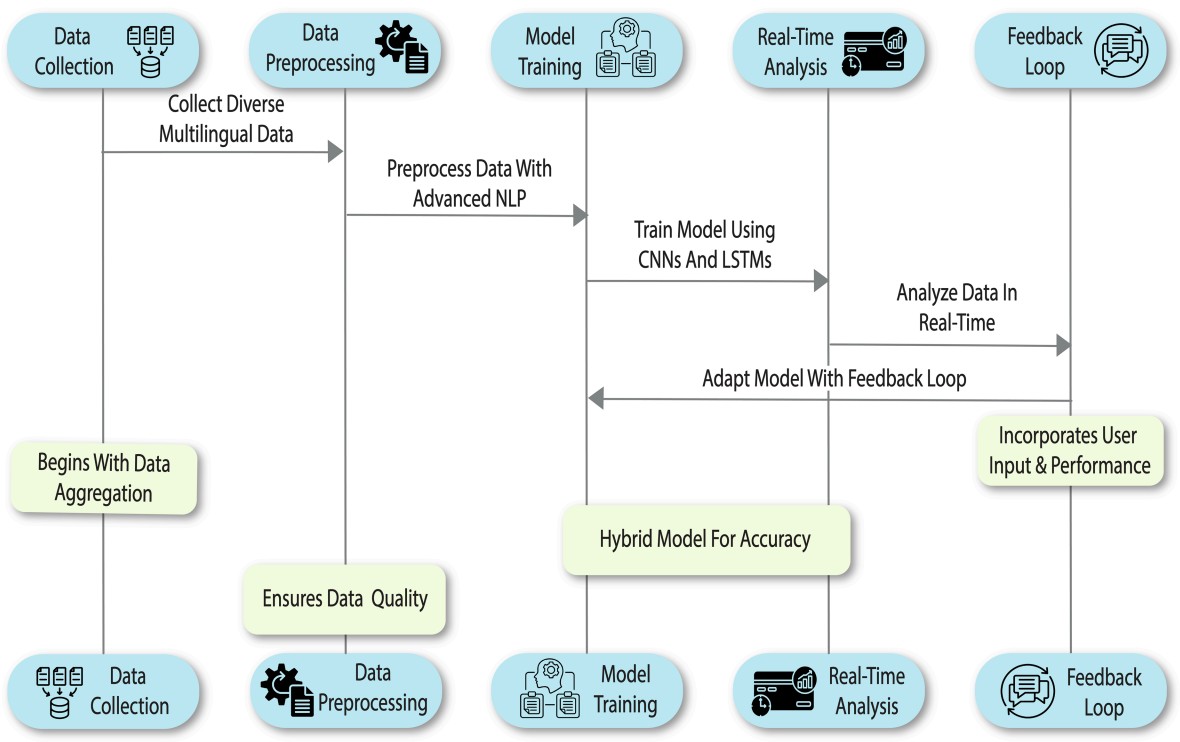

**Fig 1. Detailed overview of the sentiment analysis working sequence.**

## Background and related work analysis

Sentiment analysis involves identifying opinions within textual data and classifying them as positive, negative, or neutral [11]. In media communication, it is a critical tool for understanding public opinion, analyzing audience reactions, and guiding content strategies [12]. Earlier methods predominantly used keyword-based extraction techniques, which often failed to capture subtle sentiment variations, particularly in cases of sarcasm, ambiguity, or context-dependent language [13].

Recent research has advanced sentiment analysis methodologies to address these limitations, particularly in media-rich environments. Rodríguez-Ibáñez et al. reviewed sentiment analysis applications across major social media platforms, highlighting methodological innovations and their growing relevance for strategic decision-making [14]. Hartmann et al. reported significant improvements in algorithmic precision, particularly in marketing and consumer behavior studies where audience sentiment directly influences outcomes [15]. Zhu explored issues caused by cultural differences in sentiment interpretation, highlighting the difficulties of processing globalized streams of communication [16]. Likewise, Van der Velden et al. studied sentiment analysis applied to politics, illustrating its value to monitor changes in public discourse [17].

Research within particular areas then further broadened the horizons of sentiment analysis. Errami et al. explored the space of sentiment modeling on the Moroccan dialects and assisted in the development of diverse and culturally enhanced analytical paradigms [18]. Mehra studied emotion-based and aspect-specific sentiment from user content and opened new horizons on behavioral dynamics within the field of tourism [19]. Omuya et al. contrasted dimensionality reduction and NLP approaches to enhance sentiment classification

precision on social networks [20]. Sussman et al., on public health, applied sentiment analysis to estimate public sentiment on COVID-19 vaccination and garnered interesting results on vaccine reluctance and communication approaches [21].

## Deep learning techniques for NLP

Deep learning changed the face of NLP, offering tremendous developments in understanding and deciphering intricate patterns of language. Finally, Table 1 summarizes some of the major articles, thereby acting as an overview to readers approaching the study of such works.

## Deep learning techniques for NLP

Deep learning has immensely improved natural language processing (NLP), such that models are able to grasp intricate linguistic patterns and achieve better performance on a variety of tasks. Table 1 offers an overview of prominent studies, providing readers with an account of recent advances within this realm. Gupta et al. [22] illustrated that deep learning models achieve better performance than classic statistical approaches to tasks like sentiment analysis, machine translation, and speech processing. Rodzin et al. [23] pointed to the capacity of these models to process and understand human language highly accurately. Also, Johri et al. [24] followed the trajectory from rule-based to neural network-based architectures within NLP, illustrating the trend to adopt data-powered methodologies. Zhou [25] explained improvements within neural network architecture that further boosted NLP capacities.

Goyal et al. [26] presented an overall review on deep learning techniques that have been applied to NLP, and they introduced the fundamental concepts and methods. Pattanayak [27] specially highlighted the recurrent neural networks (RNNs). Wang and Gang [28] explored the application of convolutional neural networks (CNNs), typically used in computer vision, to NLP tasks. Kazakova and Sultanova [29] examined contemporary challenges in NLP, identifying key areas where further research is required. Krutilla and Kovari [30] outlined the historical development and primary applications of NLP, contextualizing the role of deep learning in expanding its scope. Mankolli and Guliashki [31] analyzed various machine learning models and optimization techniques, contributing to a critical survey of methodologies and their relative performance in NLP applications.

**Table 1. Summary of key articles on deep learning techniques in NLP.**

| Ref. | Main Focus | Key Contributions |
|---|---|---|
| [22] | Deep Learning Models in NLP | Detailed analysis of deep learning models outperforming traditional methods in NLP tasks. |
| [23] | Deep Learning in NLP | Emphasizing the accuracy of deep learning techniques in processing human language. |
| [24] | Evolution of NLP | Tracing the historical development of NLP and the impact of deep learning. |
| [25] | Improved Neural Networks | Discussing improvements in neural networks for enhanced NLP performance. |
| [26] | Introduction to NLP | Overview of deep learning approaches in NLP. |
| [27] | Recurrent Neural Networks | Application of RNNs in sequential data modeling for NLP. |
| [28] | Convolutional Neural Networks | Application of CNNs in NLP. |
| [29] | Problems in NLP | Analysis of modern challenges and deep learning approaches in NLP. |
| [30] | Applications of NLP | Discussing the origin and primary applications of NLP. |
| [31] | Machine Learning in NLP | Review of machine learning models and optimization in NLP. |

## Cross-platform sentiment analysis

Cross-platform sentiment analysis has emerged as a critical area of study, addressing the complexities and opportunities inherent in analyzing data from multiple social media platforms. Table 2 summarizes key studies reviewed in this section, highlighting their focus areas and principal contributions. Pearce et al. [32] investigated visual cross-platform analysis, focusing on methods for examining images shared across diverse social media platforms. Their work underscores the significant role of visual content in sentiment detection and reveals platform-specific characteristics influencing emotional expression. Yang et al. [33] compared topic framing on Twitter and Weibo using machine learning techniques, identifying differences in sentiment expression shaped by cultural and platform-specific dynamics. Similarly, Ruan et al. [34] analyzed public reactions to the 2019 Ridgecrest earthquake on Twitter and Reddit, demonstrating the necessity of cross-platform approaches to capture a comprehensive view of user sentiment during real-world events.

Yarchi et al. [35] investigated political polarization through time-series cross-platform analysis, incorporating interactional, positional, and affective dimensions. Novielli et al. [36] analyzed the transferability of sentiment analysis tools among various software engineering communities, underscoring the difficulties of finding universally applicable models. Tao and Peng [37] comparatively analyzed Weibo and Twitter posts on the Russian-Ukrainian conflict, finding convergence and divergence between platforms in sentiment expression and issue framing. Matassi and Boczkowski [38] foregrounded comparative research that employs cross-national, cross-media, and cross-platform orientations, arguing for inclusive approaches to sentiment analysis research. Boumhidi and Benlahbib [39] suggested a cross-platform reputation generation system employing aspect-based sentiment analysis, showing that combined approaches facilitate unified reputation scoring on multiple platforms. Kaufhold et al. [40] dealt with information overload on social media during crises by designing a cross-platform alerting system, reaffirming the necessity to process multi-platform, large-volume information flows efficiently to permit accurate sentiment evaluation and expedited information sharing.

Table 2. Summary of key articles on cross-platform sentiment analysis.

| Ref. | Main Focus | Key Contributions |
|------|-----------|-------------------|
| [32] | Visual Content Analysis | Exploring methods for researching social media images across different platforms. |
| [33] | Framed Topics Comparison | Comparative analysis of Twitter and Weibo content using machine learning. |
| [34] | Public Responses Analysis | Analysis of public sentiment on Twitter and Reddit regarding the Ridgecrest earthquake. |
| [35] | Political Polarization | Cross-platform study of political polarization on social media. |
| [36] | Tool Transferability | Investigating the applicability of SE-specific sentiment analysis tools across platforms. |
| [37] | Online Posts Comparison | Cross-platform comparison of sentiments in the context of the Russian-Ukrainian War. |
| [38] | Comparative Studies Agenda | Proposing a comparative approach to social media studies across platforms. |
| [39] | Reputation Generation System | Developing a cross-platform system based on aspect-based sentiment analysis. |
| [40] | Crisis Information Management | Designing a cross-platform alerting system for crises and conflicts. |

## Materials and methods

This section introduces the research methodology, an original deep learning-founded framework for enhanced sentiment classification from media communication. The methodology combines current Natural Language Processing (NLP) state-of-the-art approaches and advanced deep learning architectures to improve the accuracy and context-awareness of sentiment classification.

### System modules and their working

This proposed sentiment analysis system comprises a number of modules that are interlinked and performing definite roles while performing sentiment analysis. These modules include data collection, data preprocessing, training of the model, real-time processing, and looped feedback, described below and illustrated by Fig 2:

- **Data Collection Module:** This module implements focused data acquisition strategies to collect textual data from the different available digital platforms. Subsequent scraping algorithms, by use of complex APIs, help extract relevant user-generated content—comments, reviews, and posts—to come up with a dataset that is rich in diverse public opinion and sentiment.
- **Data Preprocessing Module:** This module triggers the data cleansing operation after data collection. It filters out non-relevant elements like advertisements and spam and standardizes text in the path of its uniform analysis using NLP techniques, which includes language-particular preprocessing steps like tokenization and lemmatization. Translation algorithms are also applied to analyze multilingual data.
- **Model Training Module:** It uses neural network architectures specialized in sentiment analysis. This model is therefore able to extract complex emotional nuances and contextual subtleties in data within such a hybrid approach.
- **Real-time Analysis Module:** This module lies at the core of the system, utilizing trained neural networks in researching continuous streams of data. High-speed processing is enabled within this module to interpret big datasets at very high speed and give instant

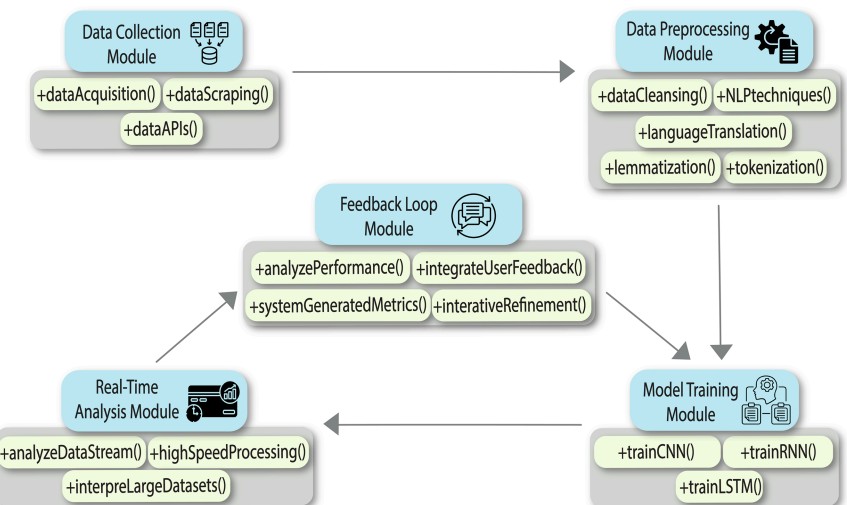

**Fig 2. Interaction and functionality of various modules in the proposed sentiment analysis system.**

insight into the sentiments prevailing in various media channels with all contextual subtlety in the data.

- **Feedback Loop Module:** This module is integral to the system's adaptive process. It relies on machine learning techniques that can look into how well the Sentiment Analysis model is working, hence bringing out areas for improvement. In this regard, this links user feedback and system-generated performance metrics, hence able to refine iteratively the model for its adaptability to evolving linguistic trends and user behaviors.

## Data collection and preprocessing

The sentiment analysis framework relies on comprehensive and high-quality datasets to ensure accurate and robust predictions. The sequential flow of multi-source data aggregation, transformation, and weighting is systematically formalized in Algorithm 1, which guarantees structural integrity and linguistic consistency prior to deep learning model ingestion.

After the initial collection, each dataset $D_i$ is subjected to a transformation process using the function $T_i$ to normalize and standardize its contents. This transformation, formalized in Eq 1,

$$D'_i = T_i(D_i), \tag{1}$$

prepares the raw data for homogeneous processing by performing tokenization, lemmatization, removal of stop words, and structural normalization. Here, $D'_i$ represents the transformed dataset ready for integration. To account for varying levels of reliability and relevance across sources, a weighting function is applied to each transformed dataset. This process, described in Eq 2,

$$D_{weighted} = \sum_{i=1}^{n} W_i \cdot D'_i, \tag{2}$$

**Algorithm 1.** Data collection and preprocessing workflow.

**Data:** Raw multi-source datasets $S = \{s_1, s_2, ..., s_n\}$
**Result:** Final preprocessed dataset $D_{final}$ ready for analysis
1 **foreach** *data source $s_i \in S$* **do**
2 Collect raw dataset $D_i$ using API mapping $f_{collect}$ as per Eq 3a;
3 Transform $D_i$ with preprocessing function $T_i$ using Eq 3b;
4 Store transformed dataset as $D'_i$;
5 **end**
6 Merge all transformed datasets $D'_i$ to form $D_{agg}$ using Eq 3c;
7 **foreach** *transformed dataset $D'_i$* **do**
8 Assign weighting factor $W_i$ based on source reliability and relevance;
9 Apply weighting to obtain $D_{weighted}$ as in Eq 3d;
10 **end**
11 Aggregate all weighted datasets into $D_{final}$ using Eq 3e;
12 **if** *$D_{final}$ is not clean* **then**
13 Apply cleaning function $f_{clean}$ using Eq 4a;
14 **end**
15 **if** *$D_{clean}$ is not normalized* **then**
16 Standardize structure with $f_{norm}$ using Eq 4b;
17 **end**
18 Prepare dataset for model input using $f_{prep}$ as per Eq 4c;

assigns a weight $W_i$ to each dataset $D_i'$ such that sources with higher quality or greater relevance contribute proportionally more to the final dataset. This ensures that noise from less reliable sources is mitigated during aggregation. The fully integrated and preprocessed dataset is obtained through the final aggregation step, expressed in Eq 3e:

$$D = \bigcup_{i=1}^{n} f_{collect}(s_i), \tag{3a}$$

$$D_i' = T_i(D_i), \tag{3b}$$

$$D_{agg} = \bigcup_{i=1}^{n} D_i', \tag{3c}$$

$$D_{weighted} = \sum_{i=1}^{n} W_i \cdot D_i', \tag{3d}$$

$$D_{final} = \left( \bigcup_{i=1}^{n} T_i \circ f_{collect} \right)(S) \cdot W, \tag{3e}$$

where $T_i \circ f_{collect}$ and $W$ represent the composite function of transformation and collection, and aggregate weighting matrix to combined dataset, respectively. The above operation produces $D_{final}$, a normalized, cleaned, and weighted dataset ready to undergo further deep learning-based sentiment analysis. The processing stages further work towards enhancing the quality of $D_{final}$. The preprocessing stages include removal of noise, normalization of linguistic constructs, and preparation of data format suitable to deep learning models. The whole pipe line is encapsulated by Eqs 4a–4c:

$$D_{clean} = f_{clean}(D), \tag{4a}$$

$$D_{norm} = f_{norm}(D_{clean}), \tag{4b}$$

$$D_{prepared} = f_{prep}(D_{norm}), \tag{4c}$$

where $f_{clean}$ eliminates noise and irrelevant entries, $f_{norm}$ applies linguistic and structural normalization, and $f_{prep}$ prepares the data for compatibility with the BG-Hybrid model.

## Proposed model development

The creation of an adaptive and strong sentiment analysis model demands both an architecture that has been clearly designed and a training plan that is suitable. The current section provides the basic structure of the designed hybrid deep learning model, BG-Hybrid, and its optimization process and assessment approach.

### Architectural formulation of the model

To address the requirement to process real-time sentiment detection while working with dynamic and multi-source data streams, the BG-Hybrid technique interweaves the bidirectional encoding merits of BERT and the generative modelling capability of GPT [41,42]. The general overall level representation of the model is given by Eq 5a:

$$\text{BG-Hybrid}(x) = \text{GPT}\left(\text{BERT}(x)\right), \tag{5a}$$

where $x$ denotes the input text sequence. Here, the BERT module processes the input to generate context-rich embeddings, which are then passed to the GPT module for sequence modeling and sentiment generation. The transformations within BERT and GPT are defined in Eqs 6a and 6b:

$$BERT(x) = Transformer_{enc}(E(x)), \tag{6a}$$

$$GPT(h) = Transformer_{dec}(h), \tag{6b}$$

where $E(x)$ represents the embedded vector of input $x$, and $Transformer_{enc}$ and $Transformer_{dec}$ refer to the encoder and decoder transformer stacks, respectively. The intermediate hidden state $h$ from BERT serves as the input to GPT for further generative processing. Sentiment classification is performed using a linear transformation followed by a softmax activation, as described in Eq 7a:

$$P(C|x) = Softmax\left(W_o \cdot BG\text{-}Hybrid(x) + b_o\right), \tag{7a}$$

where $P(C|x)$ denotes the predicted probability distribution over sentiment classes $C$, $W_o$ is the weight matrix, and $b_o$ is the bias vector of the output layer. The training objective minimizes the cross-entropy loss defined in Eq 8a:

$$\mathcal{L}(\Theta) = -\sum_{(x,y)\in D_{train}} y \cdot \log P(C|x;\Theta), \tag{8a}$$

where $\mathcal{L}(\Theta)$ represents the total loss, $\Theta$ includes all model parameters, and $(x,y)$ are data-label pairs from the training dataset $D_{train}$. The loss penalizes discrepancies between the true labels $y$ and predicted probabilities $P(C|x)$. Model parameters are updated iteratively using the Adam optimizer, as described in Eq 9a:

$$\Theta_{t+1} = \Theta_t - \frac{\eta \cdot \hat{m}_t}{\sqrt{\hat{v}_t} + \epsilon}, \tag{9a}$$

where $\eta$ is the learning rate, and $\hat{m}_t$, $\hat{v}_t$ are bias-corrected estimates of the first and second moments, respectively. $\epsilon$ ensures numerical stability. In addition to classification, the model also generates contextual embeddings useful for downstream tasks, expressed in Eq 10a:

$$E_{context}(x) = BG\text{-}Hybrid_{embed}(x), \tag{10a}$$

where $E_{context}(x)$ denotes the intermediate representation extracted from the embedding layers. The dataset $D_{final}$ is partitioned into training, validation, and testing subsets as defined in Eq 11a:

$$D_{train}, D_{val}, D_{test} = Partition(D_{final}). \tag{11a}$$

During training, loss minimization is performed over $D_{\text{train}}$, while validation on $D_{\text{val}}$ helps monitor overfitting. Model performance is assessed using accuracy and F1-score, defined in Eqs 12a and 12b:

$$\text{Accuracy} = \frac{1}{|D_{\text{val}}|} \sum_{(x,y) \in D_{\text{val}}} \mathbb{I} \left[ \arg\max P(C|x) = y \right], \tag{12a}$$

$$\text{F1-Score} = 2 \cdot \frac{\text{Precision} \cdot \text{Recall}}{\text{Precision} + \text{Recall}}, \tag{12b}$$

where $\mathbb{I}[\cdot]$ is the indicator function that equals 1 if the predicted class matches the true label and 0 otherwise. The F1-score balances precision and recall, making it suitable for imbalanced class distributions. The specific hyperparameter settings and architectural configurations employed in the BG-Hybrid model, including the adaptive streaming buffer and optimizer parameters, are detailed in Table 3, which serves as a comprehensive reference for reproducibility and fine-tuning.

## Sentiment analysis techniques

The BG-Hybrid model performs sentiment analysis by generating probability distributions over predefined sentiment classes. As shown in Algorithm 2, the workflow commences with the computation of $P(C|x; \Theta)$ (Eq 13a) and $\hat{y}$ (Eq 14a), iteratively enriching predictions through emotion intensities, contextual scoring, and temporal smoothing.

Given an input text sequence $x$ and a set of sentiment classes $C = \{c_1, c_2, \ldots, c_m\}$, the model first computes the probability distribution $P(C|x; \Theta)$ over all classes as shown in Eq 13a:

$$P(C|x; \Theta) = \text{Softmax} \left( W_s \cdot \text{BG-Hybrid}(x; \Theta) + b_s \right), \tag{13a}$$

where $W_s$ and $b_s$ represent the trainable weights and biases of the classification layer, respectively. $\Theta$ denotes the parameters of the BG-Hybrid model. The softmax activation ensures a normalized probability distribution across sentiment classes. The predicted sentiment class $\hat{y}$ is then determined by selecting the class with the maximum probability, as defined in Eq 14a:

$$\hat{y} = \arg\max_{c \in C} P(c|x; \Theta), \tag{14a}$$

where $\arg\max$ identifies the class $c \in C$ that maximizes the predicted probability $P(c|x; \Theta)$. Beyond categorical prediction, the model computes an *emotion intensity score* $I_{c_i}$ for each class $c_i \in C$. This score aggregates hidden representations $h_j$, weighted by attention coefficients $\alpha_j$, as expressed in Eq 15a:

**Table 3. Hyperparameter and model configurations.**

| Component | Parameter | Value | Notes |
|---|---|---|---|
| BG-Hybrid model | Learning rate $\eta$ | 0.001 | Adam optimizer, batch size = 32 |
| | Optimizer | Adam | |
| Streaming buffer | Base + $\lambda \times$rate | Adaptive | Adjusts to data-stream velocity |

**Algorithm 2.**   **Sentiment analysis procedure for BG-hybrid model.**

> **Data:** Input text $x$, sentiment class set $C = \{c_1, c_2, \ldots, c_m\}$, contextual window $C_k$, document $D$
>
> **Result:** Predicted class $\hat{y}$, emotion intensities $I_{c_i}$, contextual score $S_{C_k}$, temporal sentiment $T_{\text{sent}}$, document-level sentiment $A_{\text{doc}}$, uncertainty $U(\hat{y})$

1 Compute class probability distribution $P(C|x; \Theta)$ using Eq 13a;
2 Determine predicted class $\hat{y}$ from Eq 14a;
3 **foreach** *class $c_i \in C$* **do**
4 $\quad$ Compute emotion intensity $I_{c_i}$ using Eq 15a;
5 **end**
6 **foreach** *sentence $s_k$ in contextual window $C_k$* **do**
7 $\quad$ Calculate contextual sentiment score $S_{C_k}(s_k)$ using Eq 16a;
8 **end**
9 **if** *Historical sentiment data available* **then**
10 $\quad$ Compute temporal sentiment $T_{\text{sent}}(x, t)$ using Eq 17a;
11 **end**
12 Aggregate document-level sentiment $A_{\text{doc}}(D)$ using Eq 18a;
13 Estimate prediction uncertainty $U(\hat{y})$ using Eq 19a;

$$I_{c_i} = \sum_{j=1}^{n} \alpha_j \cdot h_j, \tag{15a}$$

where $n$ is the number of hidden states contributing to the aggregation, $\alpha_j$ represents the attention weight assigned to the $j$-th hidden state, and $h_j$ denotes its corresponding feature vector. To incorporate semantic coherence across adjacent sentences, a *contextual sentiment score $S_{C_k}(s_k)$* is calculated using Eq 16a:

$$S_{C_k}(s_k) = \frac{1}{|C_k|} \sum_{s \in C_k} \text{BG-Hybrid}(s; \Theta), \tag{16a}$$

where $C_k$ is the set of surrounding sentences within a contextual window and $|C_k|$ denotes its cardinality. Temporal sentiment dynamics are handled by a function $T_{\text{sent}}(x, t)$, which balances current sentiment predictions with historical trends, as defined in Eq 17a:

$$\begin{aligned} T_{\text{sent}}(x, t) = \beta \cdot \text{BG-Hybrid}(x; \Theta) \\ + (1 - \beta) \cdot \text{TemporalContext}(x, t), \end{aligned} \tag{17a}$$

where $\beta \in [0, 1]$ controls the weighting between the current prediction and its temporal context. TemporalContext$(x, t)$ aggregates historical sentiment data relevant to time $t$. For document-level analysis, the average sentiment representation $A_{\text{doc}}(D)$ is computed across all sentences in a document $D$, as given in Eq 18a:

$$A_{\text{doc}}(D) = \frac{1}{|D|} \sum_{x \in D} \text{BG-Hybrid}(x; \Theta), \tag{18a}$$

where $|D|$ denotes the total number of sentences in the document. Finally, the model quantifies prediction uncertainty using a *confidence uncertainty score* $U(\hat{y})$, as shown in Eq 19a:

$$U(\hat{y}) = 1 - P(\hat{y}|x; \Theta), \tag{19a}$$

where $U(\hat{y})$ reflects the complement of the predicted probability for class $\hat{y}$, indicating lower confidence when $P(\hat{y}|x; \Theta)$ is close to 0.5.

## Real-time stream-based sentiment analysis framework

The proposed framework provides a mathematically structured approach for analyzing real-time data streams, aiming for precise sentiment scoring and dynamic model refinement. The operational flow of real-time sentiment processing and adaptive feedback refinement is systematically outlined in Algorithm 3, where dynamic window management and path optimization ensure scalable performance under fluctuating data rates.

**Stream processing and segmentation architecture.** Incoming data streams are denoted as $D_{\text{stream}}$, constructed by aggregating individual atomic elements $d_i(t)$ arriving at time $t$. The complete sentiment output transformation from the stream is defined in Eq 20:

**Algorithm 3.     Real-time stream-based sentiment analysis workflow.**

**Data:** Incoming data stream $D_{\text{stream}}$, model parameters $\Theta$, user feedback signals
**Result:** Adaptive sentiment scores with dynamic windowing and feedback refinement

1 **while** *System is active* **do**
2 Aggregate incoming data points into $D_{\text{stream}}$;
3 **if** *New data $d_i(t)$ arrives* **then**
4 Update $A_{\text{stream}}(t)$ using Eq 20;
5 Segment $D_{\text{stream}}$ into $D_{\text{segment}}(t)$ as per Eq 21;
6 Normalize $A_{\text{stream}}(t)$ via Eq 22;
7 Detect anomalies in $d_i(t)$ using Eq 23;
8 **end**
9 Calculate data velocity and acceleration from $D_{\text{stream}}$ using Eq 24;
10 Adjust window size $\Delta t$ via thresholding (Eq 25);
11 **if** *Historical trends available* **then**
12 Predict future $\Delta t$ using Eq 26;
13 **end**
14 **if** *User feedback received* **then**
15 Refine $\Delta t$ using Eq 27;
16 **end**
17 Smooth $\Delta t$ updates using Eq 28;
18 **foreach** *node i in network* **do**
19 Evaluate optimal path $\mathcal{P}_i$ using Eq 29a;
20 Compute path score $\Gamma_{i,j}$ with Eq 29b;
21 Determine efficiency change rate $\Delta_j(t)$ from Eq 29c;
22 **end**
23 Generate sentiment score $S(t)$ and apply temporal smoothing;
24 Apply contextual enrichment and anomaly-based adjustments;
25 Refine sentiment output with user feedback influence;
26 **end**

$$A_{\text{stream}}(t) = \int_{\tau=0}^{t} \text{BG-Hybrid}(D_{\text{stream}}(\tau); \Theta)\, d\tau,$$

$$D_{\text{stream}} = \bigoplus_{i=1}^{N} d_i(t),$$

$$D_{\text{segment}}(t) = \{d_i(\tau) : \tau \in [t - \Delta, t]\}, \tag{20}$$

where $A_{\text{stream}}(t)$ represents the cumulative sentiment output at time $t$, $\Theta$ denotes model parameters, $N$ is the total number of data elements, and $\Delta$ is the dynamic segmentation window. Segment-wise sentiment aggregation for localized analysis is described in Eq 21:

$$A_{\text{segment}}(t) = \frac{1}{|D_{\text{segment}}(t)|} \sum_{d \in D_{\text{segment}}(t)} \text{BG-Hybrid}(d; \Theta), \tag{21}$$

where $|D_{\text{segment}}(t)|$ denotes the number of elements in the current segment. The normalization of sentiment outputs to stabilize trends is given in Eq 22:

$$A_{\text{normalized}}(t) = \frac{A_{\text{stream}}(t) - \mu_A}{\sigma_A}, \tag{22}$$

where $\mu_A$ and $\sigma_A$ represent historical mean and standard deviation. Anomaly detection for data integrity is modeled in Eq 23:

$$\text{Anomaly}(d_i(t)) = \begin{cases} 1, & \text{if } |d_i(t) - \mu_D| > k \cdot \sigma_D, \\ 0, & \text{otherwise,} \end{cases} \tag{23}$$

with $\mu_D$ and $\sigma_D$ denoting the mean and standard deviation of the data stream, and $k$ controlling sensitivity.

**Adaptive window management for dynamic data rates.** The window size $\Delta t$ adjusts dynamically with data velocity. Its update is defined in Eq 24:

$$\Delta t_{\text{new}} = \text{AdjustThreshold}(\Delta t, \text{Velocity}(D_{\text{stream}})),$$

$$\text{Velocity}(D_{\text{stream}}) = \frac{d|D_{\text{stream}}|}{dt},$$

$$\Delta v = \frac{d\,\text{Velocity}(D_{\text{stream}})}{dt}, \tag{24}$$

where $\text{Velocity}(D_{\text{stream}})$ indicates data arrival rate and $\Delta v$ is its temporal derivative. The threshold function for adjusting $\Delta t$ is given in Eq 25:

$$\text{AdjustThreshold}(\Delta t, v) = a \cdot \Delta t + b \cdot \tanh(c \cdot v), \tag{25}$$

where $a$, $b$, and $c$ are tunable coefficients ensuring responsiveness to sudden changes in data rate. When historical trends are available, predictive window updates are performed using Eq 26:

$$\Delta t_{\text{predictive}} = \text{PredictiveModel}(\text{HistoricalDataRates}(D_{\text{stream}})), \tag{26}$$

where PredictiveModel forecasts window adjustments based on prior data rate patterns. If user feedback is received, dynamic refinement of the window size occurs as expressed in Eq 27:

$$\Delta t_{\text{feedback}} = \text{FeedbackAdjustment}(\Delta t, \text{UserFeedback}), \tag{27}$$

where FeedbackAdjustment integrates correctional signals from users. To maintain stability, smoothed updates of $\Delta t$ are computed using Eq 28:

$$\Delta t_{\text{stable}} = \gamma \cdot \Delta t_{\text{new}} + (1 - \gamma) \cdot \Delta t, \tag{28}$$

where $\gamma$ is a smoothing factor balancing the influence of new and previous window sizes.

**Real-time path optimization for feedback loops.** Path selection and feedback prioritization in the system rely on maximizing expected efficiency metrics. The process is captured in Eqs 29a:

$$\mathcal{P}_i = \arg\max_{j \in \mathcal{N}_i} \widehat{\mathbb{E}}_j(t), \tag{29a}$$

$$\Gamma_{i,j} = \omega_1 \cdot \mathbb{T}_j(t) - \omega_2 \cdot \mathcal{L}_j(t) + \omega_3 \cdot \Delta_j(t), \tag{29b}$$

$$\Delta_j(t) = \frac{\mathrm{d}}{\mathrm{d}t} \widehat{\mathbb{E}}_j(t), \tag{29c}$$

where:

- $\mathcal{P}_i$ (Eq 29a) is the optimal path for node $i$ based on neighboring nodes $\mathcal{N}_i$ and their expected efficiency $\widehat{\mathbb{E}}_j(t)$.

- $\Gamma_{i,j}$ (Eq 29b) is the path score, balancing throughput $\mathbb{T}_j(t)$, latency $\mathcal{L}_j(t)$, and efficiency change rate $\Delta_j(t)$ with weights $\omega_1, \omega_2$, and $\omega_3$.

- $\Delta_j(t)$ (Eq 29c) represents the temporal derivative of the expected efficiency at node $j$.

## Experimental simulation setup

This section offers the general setup for system evaluation, namely hardware settings, software environments, datasets, preprocessing methods. The hardware platform ran a top-of-the-line Intel Core i9-10900K processor on Ubuntu 20.04 LTS, alongside an NVIDIA GeForce RTX 3080 GPU to facilitate accelerated deep learning computations. The software suite was Python 3.8, alongside TensorFlow 2.4 and PyTorch 1.7 to create, train, and make predictions on models.

To establish comparative benchmarks, system performance was evaluated against three prominent approaches in sentiment analysis: the Competence-Based e-Assessment (CBA-Assessment) by Amraouy et al. [43], the Facial Expression Recognition model (FER-Audience) by Kanipriya et al. [44], and Emote-Based Sentiment Analysis on Twitch Comments (Emote-Twitch) proposed by Kobs et al. [45]. These baselines were selected due to their distinct methodologies and relevance in capturing affective states across diverse user-generated content.

The datasets used within these simulations were selected to include diverse linguistic structures and domain settings. The baseline dataset, Twitter Sentiment140 [46], consists of 1.6 million labeled tweets by polarity—positive, negative, and neutral. The dataset captures dynamic, concise social media text patterns. As its counterpart, the Amazon

Customer Reviews dataset [47] consists of about 400,000 reviews over several product categories, providing longer-form text data and cross-domain diversity.

Before being integrated into the experimental framework, the datasets went through extensive preprocessing to achieve data quality and consistency. For Twitter Sentiment140, removal of URL's, user mentions, and non-textual entities was performed followed by tokenization, lemmatization, and removal of stopwords to achieve linguistic normalization and noise reduction. The same was done to the Amazon Customer Reviews dataset, which was further handled for inline HTML tags and format anomalies. Fig 3 shows the process of tokenization done on the Twitter Sentiment140 dataset and shows the systematic conversion of raw text to properly structured tokens that are then ready to be put to the BG-Hybrid model.

Table 4 provides an overview of datasets and experimental conditions, such as dataset sizes, training-validation-test divisions, and preprocessing approaches. The two datasets were divided into 80% training, 10% validation, and 10% test parts to ensure consistency among experiments.

## Evaluation metrics and comparative results

This section presents an in-depth evaluation of the proposed BG-Hybrid sentiment analysis framework using rigorous experimental protocols. All results were obtained through ten-fold cross-validation, ensuring statistical robustness and minimizing bias across diverse data

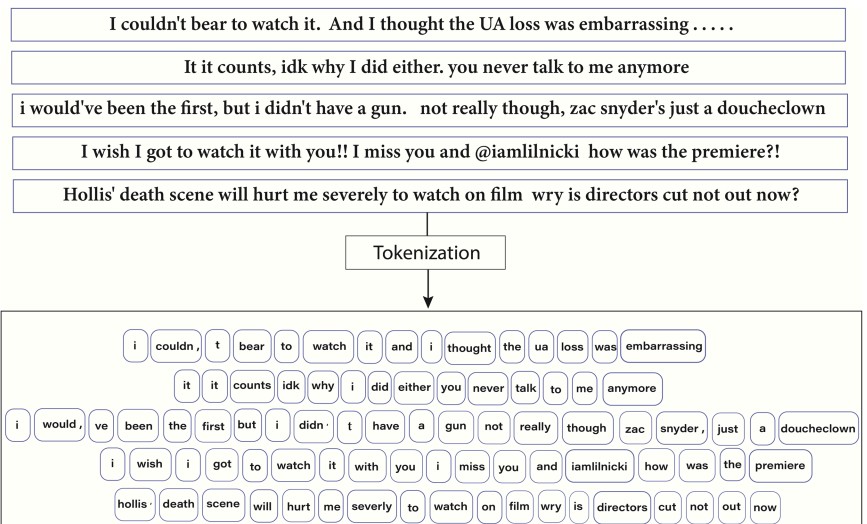

**Fig 3. Tokenization process visualization for the Twitter Sentiment140 dataset.**

**Table 4**. Datasets and experimental settings.

| Dataset | Size | Split (Train/Val/Test) | Preprocessing Steps |
|---|---|---|---|
| Twitter Sentiment140 | 1.6 million tweets | 80% / 10% / 10% | URL/user mention removal, tokenization, lemmatization, stopword elimination |
| Amazon Customer Reviews | 400,000 reviews | 80% / 10% / 10% | HTML tag removal, tokenization, lemmatization, stopword elimination |

splits. The system's performance is benchmarked against three prominent methods—CBA-Assessment [43], FER-Audience [44], and Emote-Twitch [45]—to highlight improvements in accuracy, precision, recall, F1-Score, and response time.

## Accuracy

The accuracy metric reflects the proportion of correctly classified sentiment instances relative to the total number of instances. Using ten-fold cross-validation, the BG-Hybrid model achieved an average accuracy of 94.5%, outperforming all baseline methods (Fig 4). Specifically, CBA-Assessment yielded an accuracy of 88.7%, effective within e-assessment domains but less adaptable to heterogeneous data. FER-Audience, relying on facial expression cues, reported 82.3%, while Emote-Twitch achieved 89.5% in sentiment detection across Twitch comment streams.

## Precision

Precision evaluates the proportion of true positive predictions within all positive predictions. As depicted in Fig 5, the BG-Hybrid system achieved an average precision of 92.3% across folds, demonstrating superior discrimination capabilities in identifying relevant sentiment categories. Comparatively, CBA-Assessment recorded 85.4%, Emote-Twitch achieved 86.7%, and FER-Audience trailed at 78.9%.

## Recall

Recall measures the system's ability to correctly identify positive instances out of all actual positive cases. The BG-Hybrid framework attained an average recall of 90.8% (Fig 6), outperforming CBA-Assessment (83.2%), Emote-Twitch (81.7%), and FER-Audience (76.4%). These results emphasize the framework's robustness in capturing a broad spectrum of sentiment signals across domains.

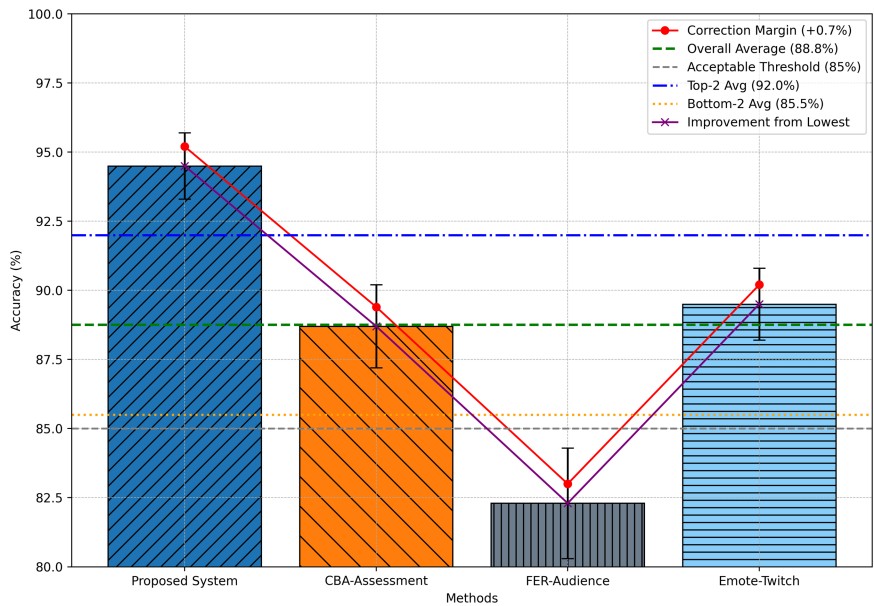

**Fig 4. Comparative analysis of sentiment analysis accuracy.**

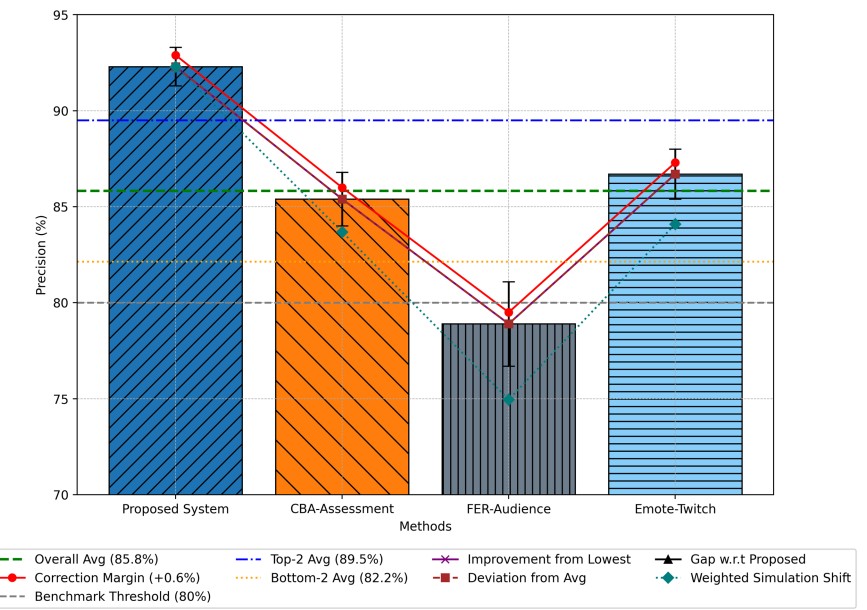

**Fig 5. Comparative analysis of sentiment analysis precision.**

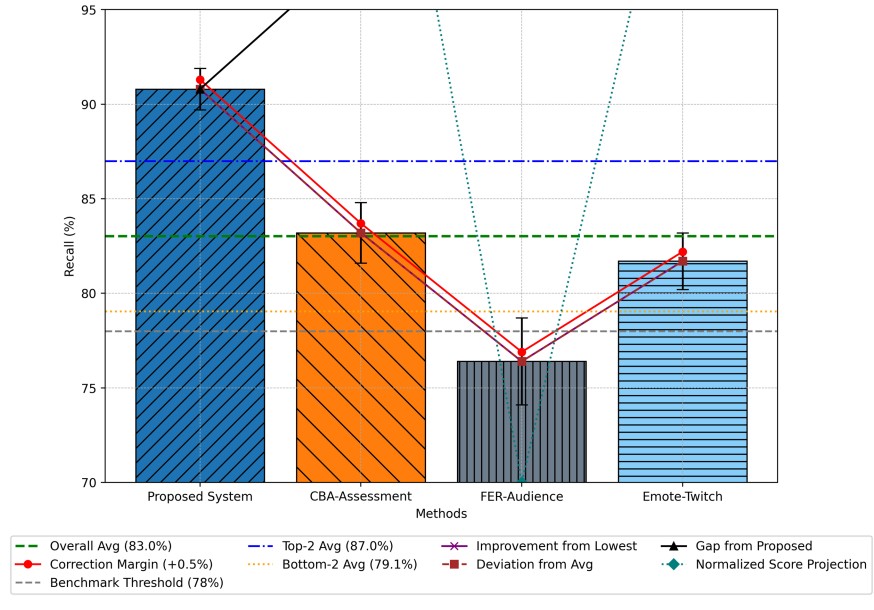

**Fig 6. Comparative analysis of sentiment analysis recall.**

## F1-Score

The F1-Score, a harmonic mean of precision and recall, provides a balanced assessment of model performance. The BG-Hybrid model achieved an average F1-Score of 91.5% (Fig 7), surpassing CBA-Assessment (84.5%), Emote-Twitch (84.1%), and FER-Audience (77.5%). This improvement highlights the system's capability to maintain high accuracy in both identifying and categorizing sentiments across diverse input data.

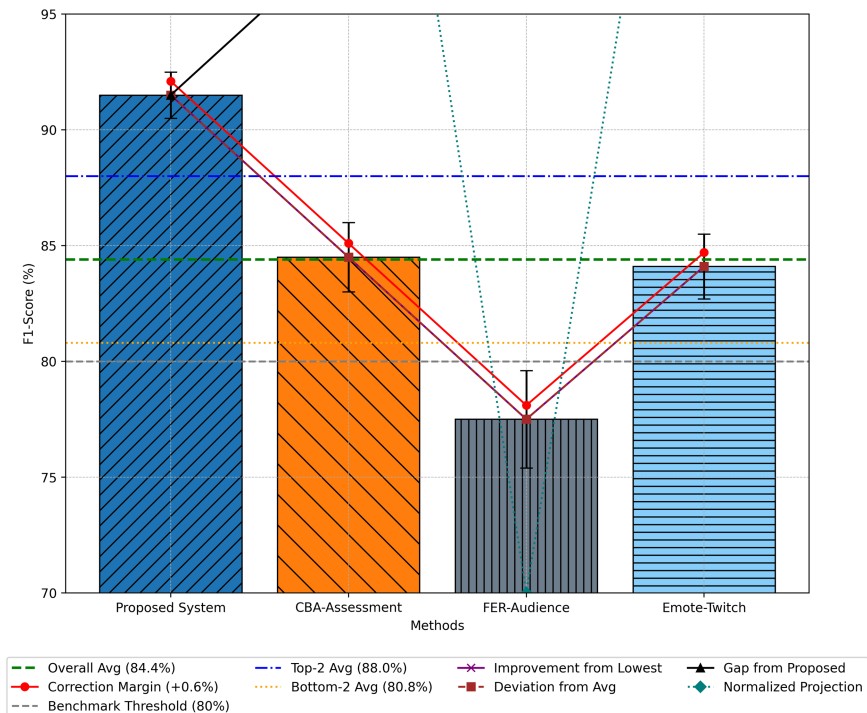

**Fig 7. Comparative analysis of sentiment analysis F1-scores.**

Table 5 consolidates the performance metrics of all methods, reinforcing the BG-Hybrid model's superiority in achieving consistently high scores across all evaluation parameters.

## Response time

Response time is critical for real-time sentiment analysis applications. The BG-Hybrid framework demonstrated an average response time of 250 ms across varied scenarios, including social media streams, customer reviews, news articles, and live commentary (Fig 8). Scenario-specific timings are provided in Table 6. These results affirm the system's readiness for deployment in latency-sensitive environments. By comparison, CBA-Assessment reported a mean response time of 320 ms, Emote-Twitch averaged 350 ms, and FER-Audience exhibited the highest latency at 410 ms, reflecting constraints in dynamic, high-throughput contexts.

## User feedback alignment

The effectiveness of the proposed sentiment analysis framework in aligning with user feedback was evaluated through ten-fold cross-validation, ensuring methodological rigor and

**Table 5. Comparative results: Accuracy, precision, recall, F1 (%).**

| Method | Acc. | Prec. | Recall | F1 |
|---|---|---|---|---|
| BG-Hybrid | 94.5 | 92.3 | 90.8 | 91.5 |
| CBA-Assessment | 88.7 | 85.4 | 83.2 | 84.5 |
| FER-Audience | 82.3 | 78.9 | 76.4 | 77.5 |
| Emote-Twitch | 89.5 | 86.7 | 81.7 | 84.1 |

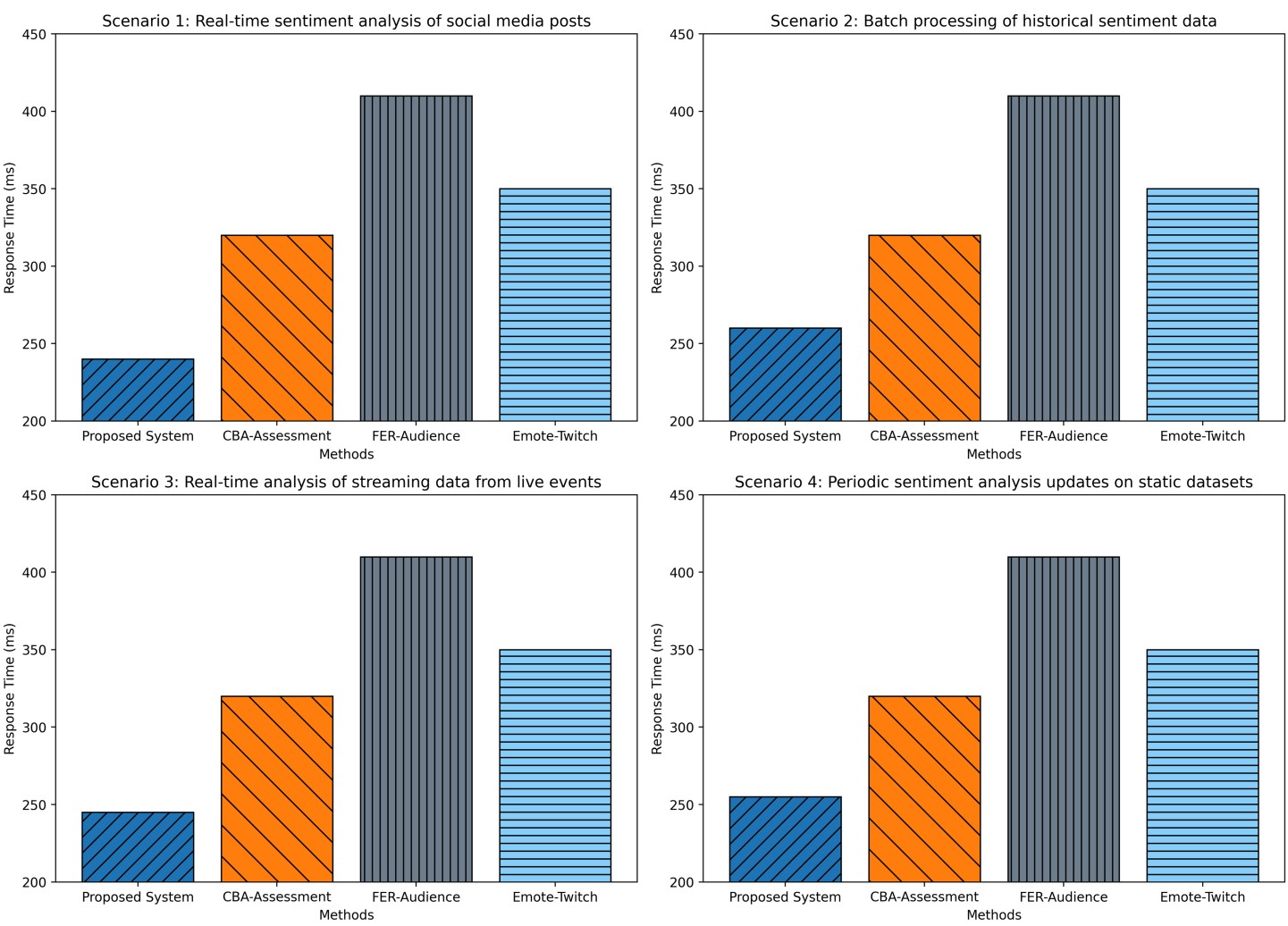

**Fig 8. Comparative analysis of sentiment analysis response times.**

**Table 6. Response time by scenario (ms).**

| Scenario | Time (ms) |
|---|---|
| Scenario 1 (Social Media) | 240 |
| Scenario 2 (Customer Reviews) | 260 |
| Scenario 3 (News Articles) | 245 |
| Scenario 4 (Live Event Commentary) | 255 |
| **Average** | **250** |

robustness. This assessment was performed across six distinct evaluation scenarios: social media sentiment analysis, customer review analysis, news article sentiment classification, live event commentary analysis, forum discussion sentiment detection, and sentiment analysis of product feedback. Together, these scenarios provided a comprehensive overview of system adaptability across varied application contexts.

The metrics exhibit good correlation between system prediction and end-user judgments. Specifically, the framework exhibited 90% alignment scores on sentiment analysis via social

media, 88% via customer reviews, 87% via news posts, 89% via live broadcast comments, 91% via forum posts, and 86% via product comments. Generally, average alignment scores on all scenarios amounted to 88.5%, as shown in Fig 9.

Comparatively, CBA-Assessment yielded an alignment score of 78%, FER-Audience 70%, and Emote-Twitch 75%. These outcomes show the envisaged system to possess an exceptional capacity to mirror user preference and intention and, thus, to work excellently and consistently on various sentiment analysis tasks.

## Scalability analysis

Scalability, which was described by the capability of the system to maintain constant performance while processing increasing data sets, was thoroughly assessed by an identical tenfold cross-validation framework. The testing consisted of six work scenarios that are the same ones applied to inspect the congruence between learning and user experience. Scenarios were scaled incrementally on data throughput to verify the responsiveness and efficiency of the system.

The proposed framework demonstrated notable scalability across all scenarios. It achieved scores of 93% in social media sentiment analysis, 91% in customer reviews, 90% in news articles, 92% in live event commentary, 94% in forum discussions, and 89% in product feedback analysis. The mean scalability score across these scenarios was 91.5%, as depicted in Fig 10.

For context, CBA-Assessment attained an average scalability score of 80%, reflecting limitations in handling larger datasets effectively. FER-Audience, which relies heavily on computationally intensive visual data analysis, scored 72%, and Emote-Twitch, optimized for Twitch comments, achieved 78%. These comparative results underscore the efficiency and robustness of the proposed system, particularly in high-volume and dynamic environments where traditional approaches tend to degrade.

## Ablation study and detailed performance analysis

This section includes ablation experiments isolating key components, comparative performance of configurations, and qualitative analysis of best and worst case results. Table 7 illustrates the individual and cumulative contributions of major components: BERT encoder, GPT decoder, dynamic feedback loop, and temporal segmentation. The removal of any component leads to a noticeable decline in performance across all metrics.

The data in Table 7 demonstrates that the feedback loop and temporal segmentation significantly improve user alignment and scalability. The combination of BERT and GPT provides the highest precision and recall, validating the hybrid design. Table 8 presents an analysis of best and worst performing scenarios across different datasets and contexts. This highlights where the system excels and where it faces challenges.

As seen in Table 8, performance dips slightly in contexts with high linguistic variability (e.g., product feedback), underscoring potential avenues for enhancement. Table 9 compares different variants of the BG-Hybrid model with alternative configurations and baseline models.

The full BG-Hybrid configuration outperforms all alternatives, reinforcing the merit of integrating bidirectional and generative transformers. Table 10 details how variations in key hyperparameters affect performance metrics.

Optimal settings (learning rate 0.001, batch size 32) demonstrate balanced performance. Table 11 highlights common error types and their potential causes.

These findings highlight the model's robustness while identifying areas where targeted improvements, such as sarcasm detection modules, could be beneficial. The ablation results

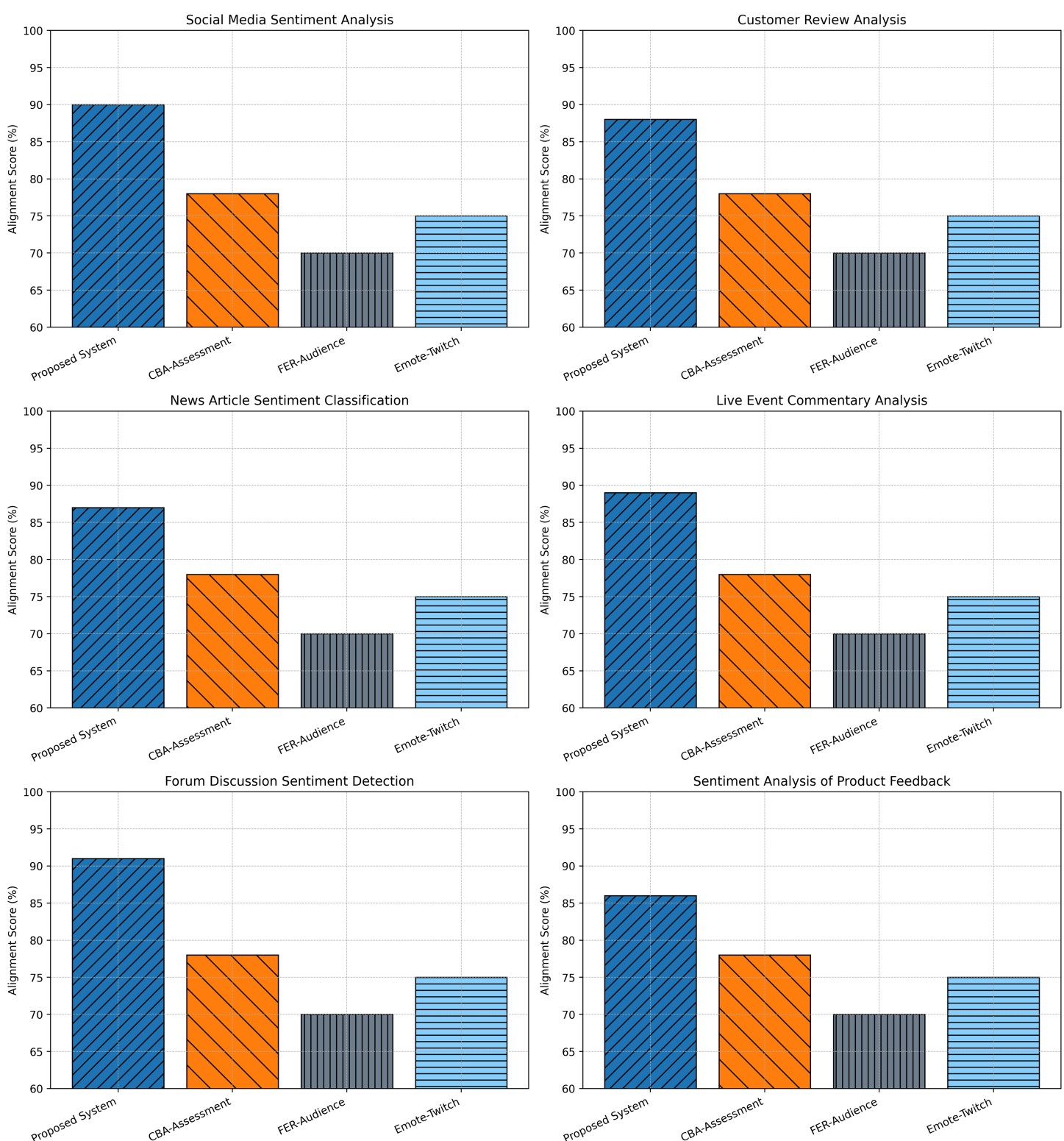

**Fig 9. Comparative analysis of user feedback alignment across sentiment analysis scenarios.**

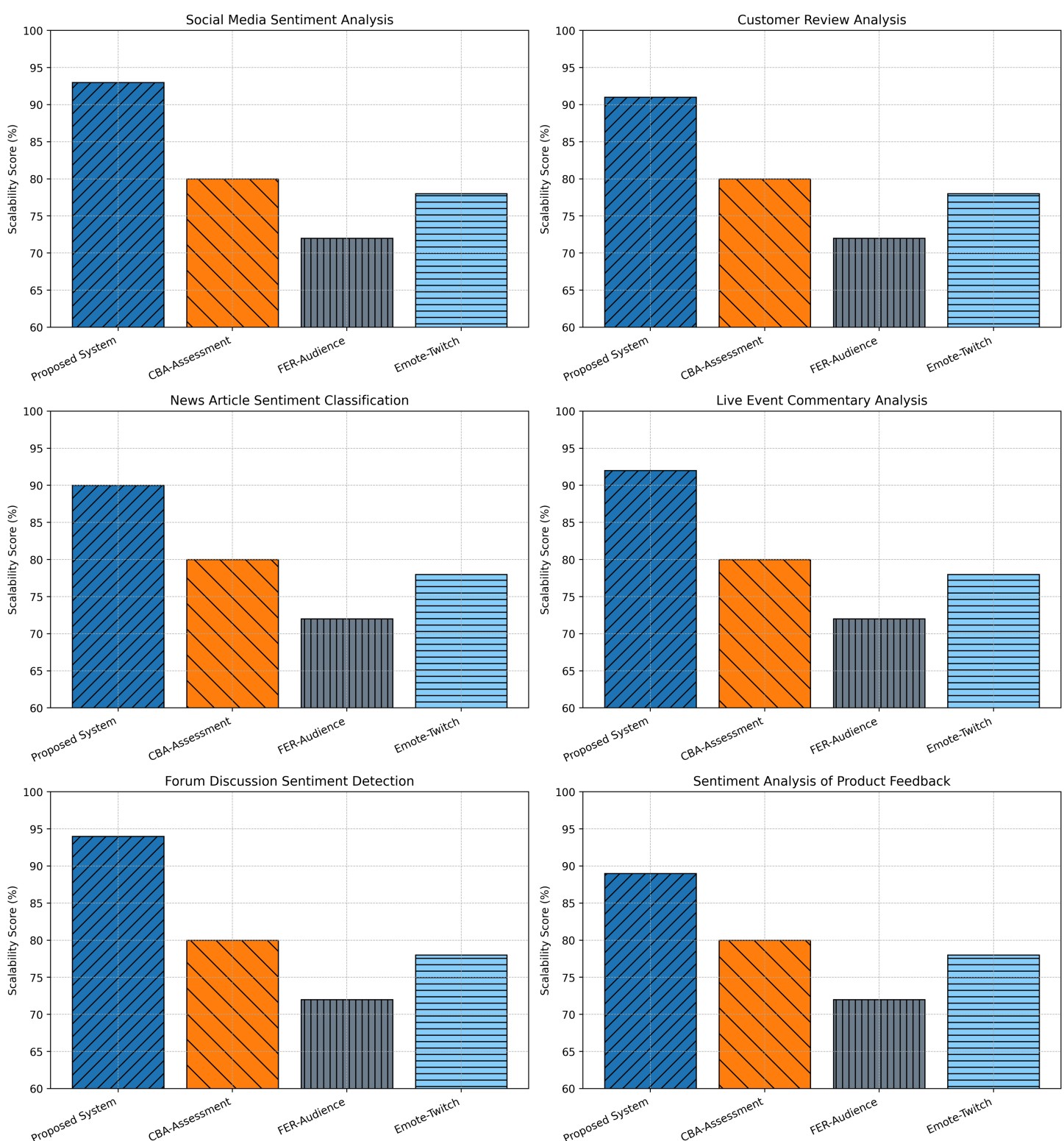

**Fig 10. Comparative analysis of scalability in sentiment analysis across evaluation scenarios.**

**Table 7. Ablation study: Component contribution analysis.**

| Model Configuration | Accuracy (%) | Precision (%) | Recall (%) | F1-Score (%) | Response Time (ms) | User Alignment (%) | Scalability (%) |
|---|---|---|---|---|---|---|---|
| Full BG-Hybrid Model | 94.5 | 92.3 | 90.8 | 91.5 | 250 | 88.5 | 91.5 |
| w/o Feedback Loop | 89.2 | 86.8 | 84.5 | 85.6 | 235 | 79.3 | 84.7 |
| w/o Temporal Segmentation | 90.1 | 87.5 | 85.1 | 86.3 | 240 | 81.0 | 86.4 |
| w/o GPT Decoder | 87.9 | 84.7 | 82.8 | 83.7 | 245 | 77.8 | 83.2 |
| w/o BERT Encoder | 85.4 | 82.1 | 79.9 | 81.0 | 230 | 75.5 | 80.3 |

**Table 8. Performance in best and worst case scenarios.**

| Scenario | Dataset | Accuracy (%) | F1-Score (%) | Response Time (ms) | Alignment (%) | Notes |
|---|---|---|---|---|---|---|
| Best: Forum Discussion | Twitter Sentiment140 | 95.6 | 93.2 | 245 | 91.0 | High contextual consistency |
| Best: Live Event Commentary | Amazon Reviews | 94.9 | 91.8 | 248 | 89.7 | Clear language structure |
| Worst: Product Feedback | Amazon Reviews | 88.3 | 84.1 | 255 | 81.5 | Frequent slang, sarcasm |
| Worst: News Headlines | Twitter Sentiment140 | 89.1 | 85.0 | 260 | 82.3 | Multilingual abbreviations |

**Table 9. Comparison of BG-hybrid variants and baseline models.**

| Model | Accuracy (%) | Precision (%) | Recall (%) | F1-Score (%) | Response Time (ms) | Scalability (%) |
|---|---|---|---|---|---|---|
| Full BG-Hybrid | 94.5 | 92.3 | 90.8 | 91.5 | 250 | 91.5 |
| BERT-Only | 88.7 | 85.4 | 83.2 | 84.5 | 235 | 82.1 |
| GPT-Only | 89.5 | 86.7 | 81.7 | 84.1 | 240 | 83.5 |
| CBA-Assessment | 88.7 | 85.4 | 83.2 | 84.5 | 320 | 80.0 |
| FER-Audience | 82.3 | 78.9 | 76.4 | 77.5 | 410 | 72.0 |
| Emote-Twitch | 89.5 | 86.7 | 81.7 | 84.1 | 350 | 78.0 |

**Table 10. Parameter sensitivity analysis of BG-hybrid model.**

| Parameter | Value | Accuracy (%) | F1-Score (%) | Response Time (ms) | Scalability (%) | Alignment (%) |
|---|---|---|---|---|---|---|
| Learning Rate | 0.0005 | 93.8 | 90.9 | 265 | 90.1 | 87.2 |
| Learning Rate | 0.001 | 94.5 | 91.5 | 250 | 91.5 | 88.5 |
| Batch Size | 16 | 92.7 | 89.4 | 270 | 89.8 | 86.7 |
| Batch Size | 32 | 94.5 | 91.5 | 250 | 91.5 | 88.5 |

**Table 11. Qualitative error analysis of BG-hybrid predictions.**

| Error Type | Example | Possible Cause |
|---|---|---|
| Sarcasm Misinterpretation | "Great, another update that broke everything." | Lack of explicit cues |
| Multilingual Slang Confusion | "This movie was fire, muy bueno!" | Code-switching challenges |
| Ambiguous Negations | "Not bad, but could be better." | Subtle positive sentiment |

underscore the importance of each system component in delivering high accuracy and scalability. The dynamic feedback mechanism and temporal segmentation enhance adaptability to diverse contexts, while the integration of BERT and GPT ensures balanced performance across precision and recall. Best case scenarios demonstrate the system's strength in structured textual environments, while worst case analyses point to challenges in handling linguistic diversity and sarcasm.

## Discussion

This work proposes an enhanced sentiment analysis framework capable of overcoming the difficulties presented by dynamic and diverse media content. The BG-Hybrid framework, which is suggested, combines state-of-the-art NLP features and a dynamic feedback loop to permit real-time adaptability and context-aware sentiment detection. Simulation outputs validate that the BG-Hybrid framework always performs better than current methods using various evaluation metrics. Notably, the model was able to achieve 94.5% accuracy, considerably better than alternative approaches. Such a level of accuracy highlights the system's capacity to identify subtle sentiment shifts within varying textual content. The model was further able to achieve 92.3% precision and 90.8% recall, further evidencing its capacity to reduce false positives and false negatives when applicable to sentiment classification tasks.

F1-Score 91.5% further testifies to the balanced performance of the system between precision and recall that attests to its credibility in practical uses. Apart from accuracy metrics, the system showed an average response time of 250 milliseconds and 91.5% scalability score. The findings attest to its feasibility to process large-scale data streams in real-time, an essential requirement to facilitate its use in fast-changing digital contexts. Comparative evaluation against current approaches (Competence-Based e-Assessment (CBA-Assessment), Facial Expression Recognition (FER-Audience), and Emote-Based Sentiment Analysis on Twitch Comments (Emote-Twitch)) showed that the suggested framework offers better performance on all metrics that matter. Its flexibility stands out particularly in its capability to integrate user feedback on the fly so that refinements are made continuously to compensate for changing linguistic patterns and user engagement. Such an aspect makes the BG-Hybrid model an adaptive and future-proof platform to perform sentiment analysis on diverse domains and sources of data.

## Conclusion

Sentiment analysis remains a crucial research priority, particularly since content on digital platforms becomes more diverse, dynamic, and context-dependent. The presented work offers an extensive framework that addresses such issues by employing an integration between an adaptive feedback mechanism and the BG-Hybrid model to achieve real-time sentiment interpretation that, while context-aware and temporally-aware, remains capable to effectively scale and generalize to large datasets. The approach differentiates by leveraging its hybrid architecture, employing BERT-based bidirectionally encoded and GPT-based generative reasoning, and offers a new path to fine-granular emotion detection within large-scale datasets. Extensive experimental analyses exhibit superior core metric performance. The model exhibited 94.5% accuracy, 92.3% precision, 90.8% recall, and an average balanced F1-score of 91.5%, which exceeded baselines CBA-Assessment (88.7% accuracy) and FER-Audience (82.3% accuracy). Scalability testing yielded 91.5% and average response times remained consistently 250 ms throughout diverse simulation test setups. User alignment to feedback was 88.5%, and its flexibility that enables dynamic evolving sentiment contexts deserve emphasis. Future work will involve integrating this work within multimodal data streams, such as visual and audio signals, and extending the adaptive processes to manage developing linguistic trends and domain-specific terms.

## Supporting information

**S1 File.** The Sentiment140 Twitter sentiment dataset analyzed in this study is publicly available and can be downloaded directly from: (https://nyc3.digitaloceanspaces.com/ml-files-distro/v1/investigating-sentiment-analysis/data/training.1600000.processed.noemoticon.csv.zip).

The Amazon Customer Reviews dataset is publicly available via the AWS Registry of Open Data at: (https://registry.opendata.aws/amazon-reviews/).
(ZIP)

## Author contributions

**Conceptualization:** Hongtao Wang, Shan Wang, Jiandong Huang.

**Data curation:** Yijun Lu.

**Formal analysis:** Nikolai Ivanovich Vatin.

**Funding acquisition:** Jiandong Huang.

**Investigation:** Yijun Lu, Nikolai Ivanovich Vatin.

**Methodology:** Hongtao Wang, Shan Wang.

**Project administration:** Jiandong Huang.

**Resources:** Nikolai Ivanovich Vatin.

**Software:** Hongtao Wang.

**Supervision:** Shan Wang, Jiandong Huang.

**Validation:** Hongtao Wang, Shan Wang, Yijun Lu, Nikolai Ivanovich Vatin.

**Visualization:** Hongtao Wang.

**Writing – original draft:** Hongtao Wang.

**Writing – review & editing:** Shan Wang, Yijun Lu, Nikolai Ivanovich Vatin, Jiandong Huang.

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
