## [Decision Letter · Decision Letter 0]

17 Jun 2025

PONE-D-25-23239Enhanced Audience Sentiment Analysis in IoT-Integrated Metaverse Media CommunicationPLOS ONE

Dear Dr. Huang,

Thank you for submitting your manuscript to PLOS ONE. After careful consideration, we feel that it has merit but does not fully meet PLOS ONE’s publication criteria as it currently stands. Therefore, we invite you to submit a revised version of the manuscript that addresses the points raised during the review process.

We look forward to receiving your revised manuscript.

Kind regards,

Hung Thanh Bui, Ph.D

Academic Editor

PLOS ONE

Journal Requirements:

This research was funded by special research topic of cultural exchange of Ministry of Education (grant number CCIPE-YXSJ-20240060), key topics of open online course guidance for undergraduate universities in Guangdong Province (grant number 2022ZXKC361), and Guangzhou Musicians Association "Music Culture research" and "Primary and secondary school music education reform project" (grant number 24GZYX003).

5. We note you have included a table to which you do not refer in the text of your manuscript. Please ensure that you refer to Table 3 in your text; if accepted, production will need this reference to link the reader to the Table.

6. We are unable to open your Supporting Information file [Main-Manuscript.tex]. Please kindly revise as necessary and re-upload.

Additional Editor Comments :

This paper presented a deep learning-based model integrating Bidirectional Encoder Representations from Transformers (BERT) with the Generative Pre-trained Transformer (GPT) for sentiment analysis. The authors did experiment on Twitter Sentiment140 and Amazon Reviews datasets and analyzed the result.

The authors should explain in detail why they used a method by integrating Bidirectional Encoder Representations from Transformers (BERT) with the Generative Pre-trained Transformer (GPT)?

They did 3 algorithms and 37 formulas, but they didn’t have any improvement on BERT and GPT, just combining all components together, it means that their framework processes and calculates all parts automatically, the authors only did on the result of some main parts. So they should present what their improvement is clearly.

They should explain how they chose all parameters in their experiments in detail

They compared with three methods; they should do more comparison with another latest studies.

Researching in NLP is going fast with many new techniques, they should refer advanced methods in recent year, for example in ACL, CL, EMNLP conferences...

Ablation study should be done on the paper.

They should show some best and worst results and analyze them in detail.

Reviewers' comments:

Reviewer's Responses to Questions

**Comments to the Author**

1. Is the manuscript technically sound, and do the data support the conclusions?

Reviewer #1: Yes

Reviewer #2: Partly

Reviewer #3: Yes

2. Has the statistical analysis been performed appropriately and rigorously? 

Reviewer #1: No

Reviewer #2: Yes

Reviewer #3: Yes

3. Have the authors made all data underlying the findings in their manuscript fully available?

Reviewer #1: Yes

Reviewer #2: Yes

Reviewer #3: Yes

4. Is the manuscript presented in an intelligible fashion and written in standard English?

Reviewer #1: Yes

Reviewer #2: Yes

Reviewer #3: Yes

5. Review Comments to the Author

Reviewer #1: 1. Although performance improvements (e.g., ~5.8 percentage points in accuracy relative to the next best starting value) are significant, the lack of p-values or confidence intervals raises questions about whether these gains hold under repeated random starts.

2. The study indicates that combining BERT-based encoding and GPT-focused generative understanding produces effective results. However, there is no clear ablation study measuring the isolated contribution of each component. Although the overall results are good, a bit ablation study would strengthen the causal link between the hybrid architecture and performance gains.

3. The average accuracy (94.5%) and F1 score (91.5%) have been determined. However, standard deviations or confidence intervals have not been determined for multiple studies or folds. Without variability measurements, it is not possible to assess whether the observed gains are stable or whether they are due to random fluctuations between different data divisions.

4. The study compares BG-Hybrid with three basic lines (CBA-Evaluation, FER-Tracker, Emote-Twitch) and indicates the absolute differences. However, no McNemar test, matched preloading, or any hypothesis test was used to confirm that these differences did not occur by chance and were statistically significant.

5. An average inference latency of 250 ms is given. However, no information is provided about the variance, percentile breakdown (e.g., 95th percentile), or repeated measurements under different loads. For real-time systems, tail‐latency can be more critical information than the average.

Reviewer #2: This paper presents a deep learning-based model integrating Bidirectional Encoder Representations from Transformers (BERT) with the Generative Pre-trained Transformer (GPT) for sentiment analysis. This paper addresses very pertinent research problem. Following are some main points/suggestions based on the overall insight of the paper:

• The language of the manuscript should be improved significantly like typos, alignment, use of complex sentences like “This section details the system analysis of the sentiment analysis system we proposed 367 and gives an evaluation of its performance under several simulations”.

• There are lot of hyper parameters like a, b, and c in equation 27 but how did you choose their value, it should be discussed in detail?

• Perform the comparative analysis with the recent transformer-based deep learning approaches.

• There are too much mathematical equations. Equations 7, 8, 9, 10 and some others like these are very common concepts and there is no need to define them as equation.

• Please present confusion matrix of the prediction result.

• The multilingual capability of the model depends on the translation of text which will slow down the approach. Furthermore, accuracy will also depend on the accuracy of translation service. How this makes the presented model a generic multilingual approach.

Reviewer #3: The manuscript demonstrates notable ambition and technical effort, with well-documented implementation and thorough empirical testing. However, some improvements in clarity, methodological transparency, and contextualization within prior literature are essential for publication readiness.

1. While the manuscript demonstrates commendable technical depth, the volume of equations appears excessive for the applied nature of the study. Many of the mathematical formulations—particularly those related to stream segmentation, feedback adjustment, and sentiment scoring—restate established concepts without clear empirical justification or practical interpretation. This level of formalism risks overshadowing the real-world applicability of the proposed system. A more balanced approach would involve streamlining the mathematical content and focusing instead on how the system operates in practice, its deployment challenges, and its impact in real-time IoT-Metaverse scenarios. Reducing equation density and emphasizing applied insights would improve clarity and better align the paper with its intended audience.

2. The manuscript often uses overly dense or jargon-heavy phrasing (e.g., “contextual sentiment score SCk,” “temporal segmentation strategy dynamically adapting to data stream velocities”), which impairs readability. Simplifying sentence structure without losing technical depth would improve accessibility.

Several sections (e.g., discussion of modules, performance comparisons) repeat similar points with only slight variation. This could be condensed to enhance flow.

3. From the model implementation perspective, although equations are thoroughly included, key architectural specifics of the BG-Hybrid model (e.g., the number of layers, embedding size, and fusion method between BERT and GPT outputs) are missing. Add a table of model hyperparameters and implementation details. Clarify how weights and feedback mechanisms are operationalized and validated.

4. The related work section is extensive but lacks critical engagement. It reads more like a catalog of previous studies, which is usually the conventional way of writing technical papers. However, this section can benefit from a serious analytical synthesis. Recent advancements in sentiment analysis, including those utilizing transformer ensembles, knowledge-aware modeling, and multilingual domain adaptation, are either absent or lightly addressed. Highlight specifically how the BG-Hybrid model differs from or improves upon models like RoBERTa, XLNet, or other BERT-GPT fusion attempts. Discuss cross-lingual and cross-modal sentiment analysis literature for broader grounding.

5. While the performance benchmarks are impressive, there is no ablation study showing the contribution of individual components (e.g., BERT-only vs. BG-Hybrid, effect of feedback module, effect of adaptive windowing). Include at least an additional baseline comparison (e.g., fine-tuned RoBERTa, XLNet, or DistilBERT). Justify the baseline choices.

Overall, this manuscript presents a promising and well-structured sentiment analysis framework suited for real-time, heterogeneous data environments like the IoT-enabled Metaverse. The hybrid deep learning architecture and its integration with feedback-driven adaptation are notable strengths. However, to meet the publication standards, the following are essential:

• Improve clarity and reduce redundancy

• Expand on methodological transparency

• Deepen engagement with related literature

• Add comparisons to more competitive baselines

• Less theoretical and more practical outlook

With these revisions, the paper would make a valuable contribution to the field of AI-driven media sentiment analysis in complex, connected environments.

6. PLOS authors have the option to publish the peer review history of their article (what does this mean?). If published, this will include your full peer review and any attached files.

Reviewer #1: No

Reviewer #2: No

Reviewer #3: **Yes: **Fatima Habib

---

## [Author Response · Author response to Decision Letter 1]

31 Jul 2025

Dear Editor‑in‑Chief,

Thank you for overseeing the review of our manuscript (PONE-D-25-23239). We have carefully addressed all reviewer and Associate Editor comments in the attached “Response to Reviewers” PDF. All manuscript changes are highlighted in the “Highlighted Manuscript” file. We appreciate your consideration and look forward to your decision.

Thank you.

---

## [Decision Letter · Decision Letter 1]

27 Aug 2025

Enhanced Audience Sentiment Analysis in IoT-Integrated Metaverse Media Communication

PONE-D-25-23239R1

Dear Dr. Huang,

We’re pleased to inform you that your manuscript has been judged scientifically suitable for publication and will be formally accepted for publication once it meets all outstanding technical requirements.

Kind regards,

Hung Thanh Bui, Ph.D

Academic Editor

PLOS ONE

Additional Editor Comments (optional):

All comments are addressed.

Reviewers' comments:

Reviewer's Responses to Questions

**Comments to the Author**

1. If the authors have adequately addressed your comments raised in a previous round of review and you feel that this manuscript is now acceptable for publication, you may indicate that here to bypass the “Comments to the Author” section, enter your conflict of interest statement in the “Confidential to Editor” section, and submit your "Accept" recommendation.

Reviewer #1: All comments have been addressed

Reviewer #3: All comments have been addressed

2. Is the manuscript technically sound, and do the data support the conclusions?

Reviewer #1: (No Response)

Reviewer #3: (No Response)

3. Has the statistical analysis been performed appropriately and rigorously? 

Reviewer #1: (No Response)

Reviewer #3: (No Response)

4. Have the authors made all data underlying the findings in their manuscript fully available?

Reviewer #1: (No Response)

Reviewer #3: (No Response)

5. Is the manuscript presented in an intelligible fashion and written in standard English?

Reviewer #1: (No Response)

Reviewer #3: (No Response)

6. Review Comments to the Author

Reviewer #1: The authors have responded satisfactorily to my comments and made the necessary additions and corrections.

Reviewer #3: (No Response)

7. PLOS authors have the option to publish the peer review history of their article (what does this mean?). If published, this will include your full peer review and any attached files.

Reviewer #1: No

Reviewer #3: **Yes: **Fatima Habib

---

## [Editor Report · Acceptance letter]

PONE-D-25-23239R1

PLOS ONE

Dear Dr. Huang,

I'm pleased to inform you that your manuscript has been deemed suitable for publication in PLOS ONE. Congratulations! Your manuscript is now being handed over to our production team.

Kind regards,

on behalf of

Dr. Hung Thanh Bui

Academic Editor

PLOS ONE